# Pixel-Perfect Depth
# with Semantics-Prompted Diffusion Transformers

**Gangwei Xu**[1,2]* **Haotong Lin**[3]* **Hongcheng Luo**[2] **Xianqi Wang**[1] **Jingfeng Yao**[1]

**Lianghui Zhu**[1] **Yuechuan Pu**[2] **Cheng Chi**[2] **Haiyang Sun**[2]† **Bing Wang**[2]

**Guang Chen**[2] **Hangjun Ye**[2] **Sida Peng**[3] **Xin Yang**[1]†✉

[1]Huazhong University of Science and Technology  [2]Xiaomi EV  [3]Zhejiang University
`https://pixel-perfect-depth.github.io`

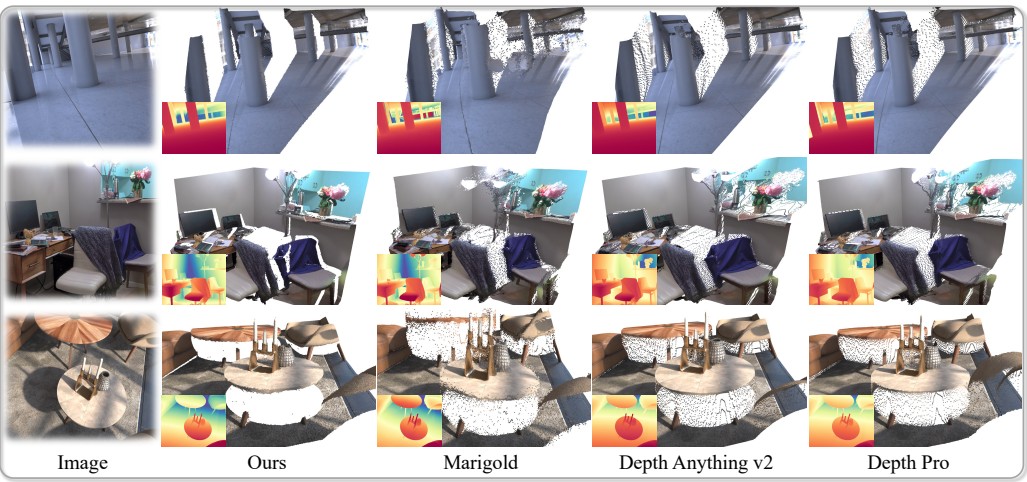

Figure 1: We present **Pixel-Perfect Depth**, a monocular depth estimation model with pixel-space diffusion transformers. Compared to existing discriminative [82, 4] and generative [34] models, its estimated depth maps can produce high-quality, flying-pixel-free point clouds.

## Abstract

This paper presents **Pixel-Perfect Depth**, a monocular depth estimation model based on pixel-space diffusion generation that produces high-quality, flying-pixel-free point clouds from estimated depth maps. Current generative depth estimation models fine-tune Stable Diffusion and achieve impressive performance. However, they require a VAE to compress depth maps into the latent space, which inevitably introduces *flying pixels* at edges and details. Our model addresses this challenge by directly performing diffusion generation in the pixel space, avoiding VAE-induced artifacts. To overcome the high complexity associated with pixel-space generation, we introduce two novel designs: 1) **Semantics-Prompted Diffusion Transformers (SP-DiT)**, which incorporate semantic representations from vision foundation models into DiT to prompt the diffusion process, thereby preserving global semantic consistency while enhancing fine-grained visual details; and 2) **Cascade DiT Design** that progressively increases the number of tokens to further enhance efficiency and accuracy. Our model achieves the best performance among all published generative models across five benchmarks, and significantly outperforms all other models in edge-aware point cloud evaluation.

---

* Equal contribution, † Project leader, ✉ Corresponding author.

39th Conference on Neural Information Processing Systems (NeurIPS 2025).

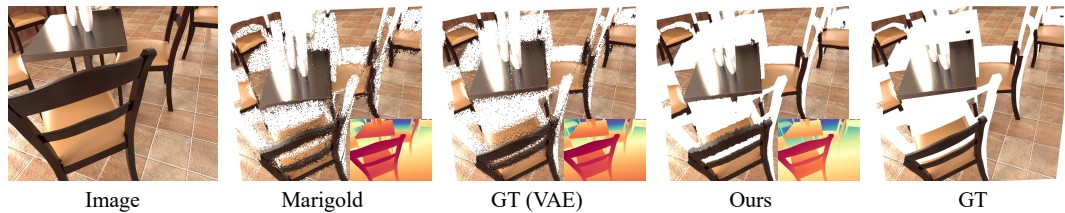

| Image | Marigold | GT (VAE) | Ours | GT |

Figure 2: **Qualitative comparisons**. GT(VAE) denotes the ground truth depth map after VAE reconstruction. Existing generative models [34] use a VAE to compress inputs into the latent space, inevitably introducing *flying pixels* at edges and details. In contrast, our model directly performs diffusion in pixel space, avoiding these issues. Depth maps are visualized on the point clouds.

# 1  Introduction

Monocular depth estimation (MDE) is a fundamental task with a wide range of downstream applications, such as 3D reconstruction, novel view synthesis, and robotic manipulation. Due to its significance, a large number of depth estimation models [34, 81, 82, 88] have emerged recently. These models achieve high-quality results in most zero-shot scenarios or regions, but suffer from *flying pixels* around object boundaries and fine details when converted into point clouds [39], as shown in Figure 1 and 5, which limits their practical applications in tasks such as free-viewpoint broadcast, robotic manipulation, and immersive content creation.

Current models suffer from the *flying pixels* problem due to different reasons. For discriminative models [82, 4, 88, 30], *flying pixels* mainly arise from their tendency to output an intermediate (*average*) depth value between the foreground and background at depth-discontinuous edges, in order to minimize regression loss. In contrast, generative models [34, 17, 23] bypass direct regression by modeling pixel-wise depth distributions, allowing them to preserve sharp edges and recover fine structures more faithfully. However, current generative depth models typically fine-tune Stable Diffusion [51] for depth estimation, which requires a Variational Autoencoder (VAE) to compress depth maps into a latent space. This compression inevitably leads to the loss of edge sharpness and structural fidelity, resulting in a significant number of *flying pixels*, as shown in Figure 2.

A trivial solution could be training a diffusion-based monocular depth model in pixel space, bypassing the use of a VAE. However, we find this highly challenging, due to the increased complexity and instability of modeling both global semantic consistency and fine-grained visual details, leading to extremely low-quality depth predictions (Table 2 and Figure 6). To further investigate this limitation, we examine prior studies on high-resolution image generation. Several works [29, 61, 94], through signal-to-noise ratio (SNR) analysis, have pointed out that adding noise with higher intensity is more likely to disrupt the global structures or low-frequency components of high-resolution images, thereby improving generation. This reveals that the primary difficulty in high-resolution pixel-space generation lies in effectively perceiving and modeling global image structures.

In this paper, we present **Pixel-Perfect Depth**, a framework for high-quality and flying-pixel-free monocular depth estimation using pixel-space diffusion transformers. Recognizing that the major difficulty in high-resolution pixel-space generation lies in perceiving and modeling global image structures. To address this challenge, we propose the **Semantics-Prompted Diffusion Transformers** (**SP-DiT**) that incorporate high-level semantic representations into the diffusion process to enhance the model's ability to preserve global structures and semantic coherence. Equipped with SP-DiT, our model can more effectively preserve global semantic consistency while generating fine-grained visual details in high-resolution pixel space. However, the semantic representations obtained from vision foundation models [44, 82, 65, 24] often do not align well with the internal representations of DiT, leading to training instability and convergence issues. To address this, we introduce a simple yet effective regularization technique for semantic representations, which ensures stable training and facilitates convergence to desirable solutions. As shown in Table 2 and Figure 6, SP-DiT significantly improves overall performance, with up to a 78% gain on the NYUv2 [58] AbsRel metric.

Furthermore, we introduce the **Cascade DiT Design** (Cas-DiT), an efficient architecture for diffusion transformers. We find that in diffusion transformers, the early blocks are primarily responsible for capturing and generating global or low-frequency structures, while the later blocks focus on

generating high-frequency details. Based on this insight, Cas-DiT adopts a progressive patch size strategy: larger patch size is used in the early DiT blocks to reduce the number of tokens and facilitate global image structure modeling; in the later DiT blocks, we increase the number of tokens, which is equivalent to using a smaller patch size, allowing the model to focus on the generation of fine-grained spatial details. This coarse-to-fine cascaded design not only significantly reduces computational costs and improves efficiency, but also delivers substantial improvements in accuracy.

We highlight the main contributions of this paper below:

- We present **Pixel-Perfect Depth**, a monocular depth estimation model with pixel-space diffusion generation, capable of producing flying-pixel-free point clouds from estimated depth maps.

- We introduce **Semantics-Prompted DiT**, which integrates normalized semantic representations into the DiT to effectively preserve global semantic consistency while enhancing fine-grained visual details. This significantly boosts overall performance. We further propose a novel **Cascade DiT Design** to enhance the efficiency and accuracy of our model.

- Our model achieves the best performance across five benchmarks among all published generative depth estimation models.

- We introduce an edge-aware point cloud evaluation metric, which effectively assesses *flying pixels* at edges. Our model significantly outperforms previous models in this evaluation.

## 2 Related Work

### 2.1 Monocular Depth Estimation

Depth estimation can be broadly categorized into monocular [82, 69], stereo [75, 72, 77, 76, 22, 8, 7, 9], and sparse depth completion [41] methods. Early monocular depth estimation methods relied primarily on manually designed features [52, 28]. The advent of neural networks revolutionized the field, though initial approaches [15, 14] struggled with cross-dataset generalization. To address this limitation, scale-invariant and relative loss [49] are introduced, enabling multi-dataset [36, 86, 10, 74, 71, 68, 64, 73, 50, 40] training. Recent methods focus on improving the generalization ability [82, 4, 66, 67], depth consistency [80, 6, 31, 33], and metric scale [3, 37, 38, 88, 21, 89, 30, 46, 41] of depth estimation. These methods converge towards using transformer-based architectures [48]. Concurrent works [69, 70, 79] explore point cloud representations to improve depth estimation performance. Several recent methods [32, 12, 55, 53, 54, 93] have attempted to use diffusion models for metric depth estimation. In contrast, our method focuses on relative depth and demonstrates improved generalization and fine-grained detail across a wide range of real-world scenes. Furthermore, our model significantly differs from these methods by introducing Semantics-Prompted DiT, which incorporates pretrained high-level semantic representations into the diffusion process, greatly enhancing performance.

More recently, [34] brought the new insight to the field by fine-tuning pretrained Stable Diffusion [51] for depth estimation, which demonstrated impressive zero-shot capabilities for relative depth. The following works [23, 20, 60, 92, 2] attempt to improve its performance and inference speed. However, they are all based on the latent diffusion model [51], which is trained in the latent space and requires a VAE to compress the depth map into a latent space. We focus on a pixel-space diffusion model that is trained directly in the pixel space without requiring any VAE.

### 2.2 Diffusion Generative Models

Diffusion generative models [25, 59, 45, 90, 84, 85, 96] have demonstrated impressive results in image and video generation. Early approaches [25, 27, 26] such as DDPM [25] operate directly in the pixel space, enabling high-fidelity generation but incurring significant computational costs, especially at high resolutions. To address this limitation, Latent Diffusion Models perform the diffusion process in a lower-dimensional latent space obtained via a VAE, as popularized by Stable Diffusion [51]. This design significantly improves training and inference efficiency and has been widely adopted in recent works [16, 85, 90, 95, 35, 47, 83].

Diffusion models for monocular depth estimation typically follow a similar trend. For instance, Marigold [34] and its follow-ups [23, 20] fine-tune pretrained Stable Diffusion [51] models for depth

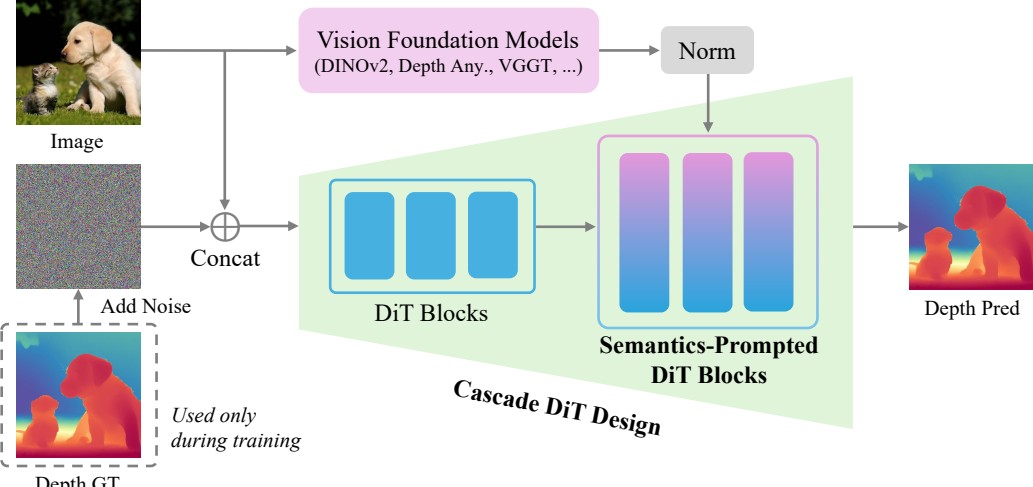

Figure 3: **Overview of Pixel-Perfect Depth.** Given an input image, we concatenate it with noise and feed it into the proposed Cascade DiT. Meanwhile, the image is also processed by a pretrained encoder from Vision Foundation Models to extract high-level semantics, forming our Semantics-Prompted DiT. We perform diffusion generation directly in pixel space without using any VAE.

estimation, benefiting from fast convergence and strong priors learned from large-scale datasets. However, the VAE's latent compression leads to *flying pixels* in the resulting point clouds. In contrast, pixel-space diffusion avoids such artifacts but remains computationally intensive and slow to converge at high resolutions. To address this, we propose Semantics-Prompted DiT and Cascade DiT Design, which enables efficient high-resolution depth estimation without latent compression.

## 3 Method

### 3.1 Pixel-Perfect Depth

Given an input image, our goal is to estimate a pixel-perfect depth map that is free of *flying pixels* when converted to point clouds. Existing models [34, 17, 23, 82, 4] often suffer from *flying pixels* due to their inherent modeling paradigms. Discriminative models tend to smooth object edges and blur fine details because of their mean-prediction bias, which results in noticeable *flying pixels* in the reconstructed point clouds. Generative models, in theory, can better capture the multi-modal depth distribution at object edges. However, current generative models typically fine-tune Stable Diffusion [51] for depth estimation, relying on its strong image priors. This requires compressing the depth map into a latent space via a VAE, inevitably causing *flying pixels*.

To unleash the potential of generative models for depth estimation, we propose **Pixel-Perfect Depth** that performs diffusion directly in the pixel space instead of the latent space. It allows us to directly model the pixel-wise distribution of depth, such as the discontinuities at object edges. However, training a generative diffusion model directly in the high-resolution pixel space (*e.g.*, 1024×768) is computationally demanding and hard to optimize. To overcome these challenges, we introduce Semantics-Prompted DiT and Cascaded DiT Design, detailed in the following sections.

### 3.2 Generative Formulation

We adopt Flow Matching [42, 43, 1] as the generative core of our depth estimation framework. Flow Matching learns a continuous transformation from Gaussian noise to a data sample via a first-order Ordinary Differential Equation (ODE). In our case, we model the transformation from Gaussian noise to a depth sample. Specifically, given a clean depth sample $\mathbf{x}_0 \sim \mathcal{D}$ and Gaussian noise $\mathbf{x}_1 \sim \mathcal{N}(0, 1)$, we define an interpolated sample at continuous time $t \in [0, 1]$ as:

$$\mathbf{x}_t = t \cdot \mathbf{x}_1 + (1 - t) \cdot \mathbf{x}_0. \tag{1}$$

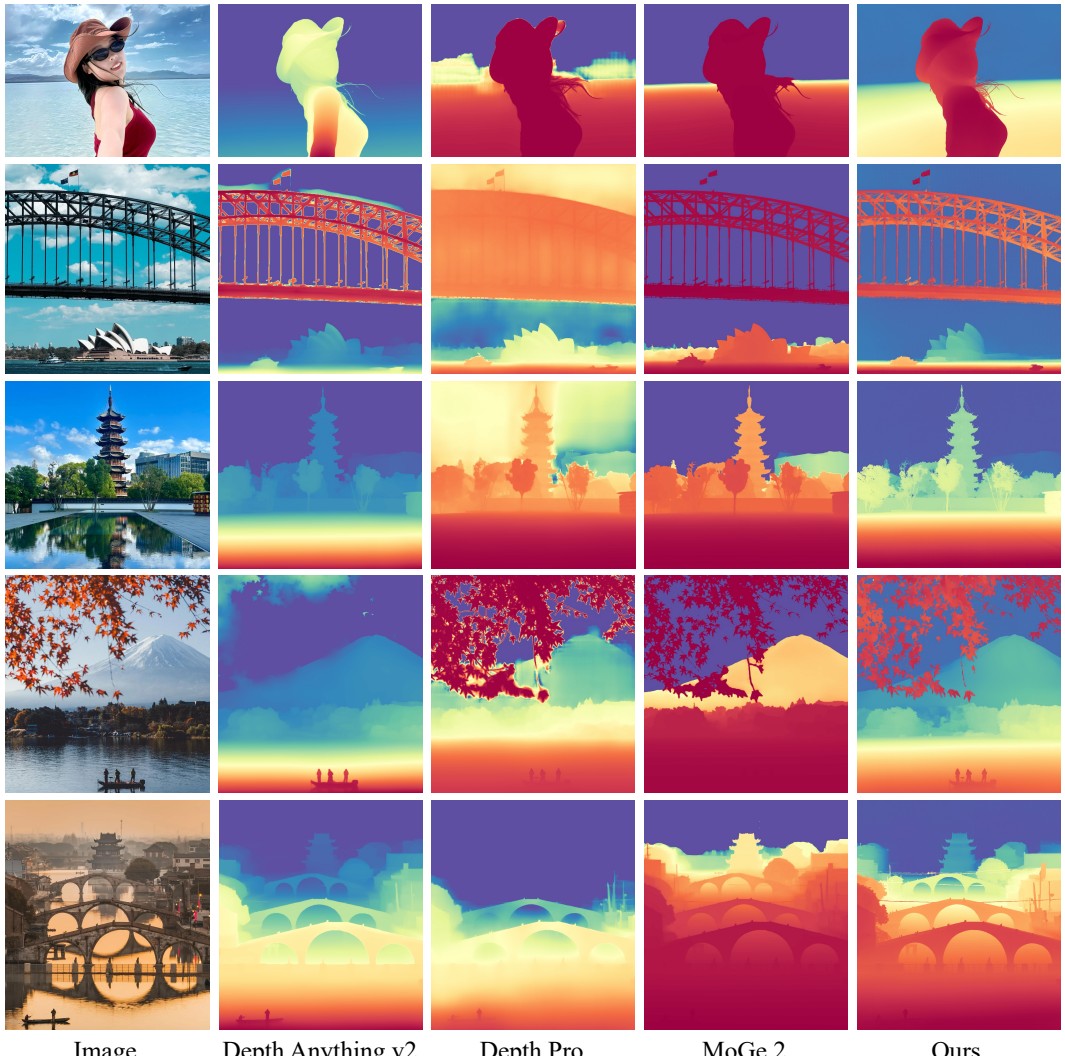

| Image | Depth Anything v2 | Depth Pro | MoGe 2 | Ours |

Figure 4: **Comparison with existing depth foundation models on open-world images.** Our model preserves more fine-grained details than Depth Anything v2 [82] and MoGe 2 [70], while demonstrating significantly higher robustness compared to Depth Pro [4].

This defines a velocity field:

$$\mathbf{v}_t = \frac{d\mathbf{x}_t}{dt} = \mathbf{x}_1 - \mathbf{x}_0, \tag{2}$$

which describes the direction from clean data to noise. Our model $\mathbf{v}_\theta(\mathbf{x}_t, t, \mathbf{c})$ is trained to predict the velocity field, based on the current noisy sample $\mathbf{x}_t$, the time step $t$, and the input image $\mathbf{c}$. The training objective is the mean squared error (MSE) between the predicted and true velocity:

$$\mathcal{L}_{\text{velocity}(\theta)} = \mathbb{E}_{\mathbf{x}_0, \mathbf{x}_1, t} \left[ \|\mathbf{v}_\theta(\mathbf{x}_t, t, \mathbf{c}) - \mathbf{v}_t\|^2 \right]. \tag{3}$$

At inference, we start from noise $\mathbf{x}_1$ and solve the ODE by discretizing the time interval $[0, 1]$ into steps $t_i$, iteratively updating the depth sample as follows:

$$\mathbf{x}_{t_{i-1}} = \mathbf{x}_{t_i} + \mathbf{v}_\theta(\mathbf{x}_{t_i}, t_i, \mathbf{c})(t_{i-1} - t_i), \tag{4}$$

where $t_i$ decreases from 1 to 0, gradually transforming the initial noise $\mathbf{x}_1$ into the depth sample $\mathbf{x}_0$.

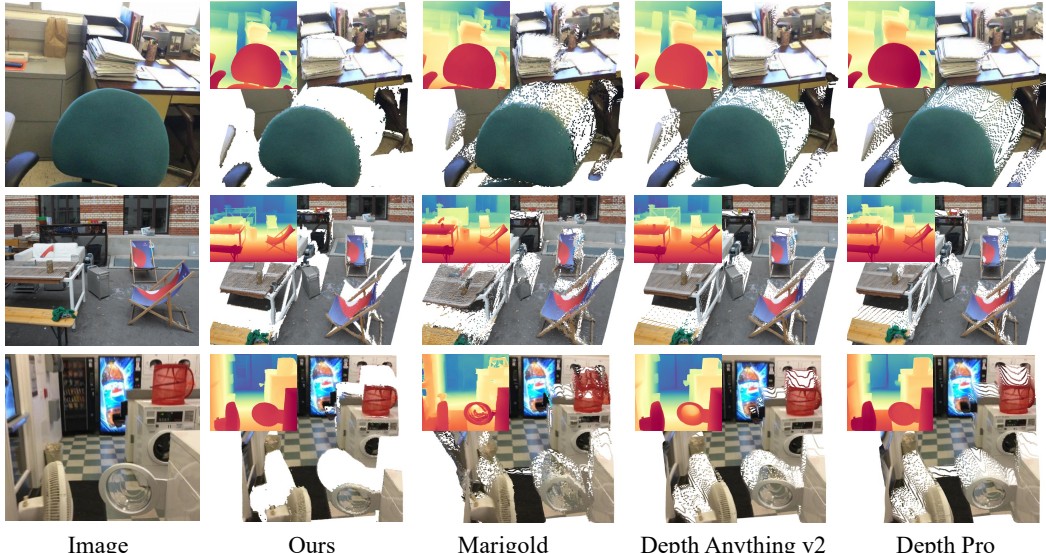

| Image | Ours | Marigold | Depth Anything v2 | Depth Pro |

Figure 5: **Qualitative point cloud results in complex scenes.** Our model produces significantly fewer *flying pixels* compared to other depth estimation models [34, 82, 4], with depth maps overlaid on the point clouds for visualization.

## 3.3 Semantics-Prompted Diffusion Transformers

Our Semantics-Prompted DiT builds on DiT [45] for its simplicity, scalability, and strong performance in generative modeling. Unlike previous depth estimation models such as Depth Anything v2 [82] and Marigold [34], our architecture is purely transformer-based, without any convolutional layers. By integrating high-level semantic representations, SP-DiT enables our model to preserve global semantic consistency while enhancing fine-grained visual details, without sacrificing the simplicity and scalability of DiT.

Specifically, given the interpolated noise sample $\mathbf{x}_t$ and the corresponding image $\mathbf{c}$, we first concatenate them into a single input: $\mathbf{a}_t = \mathbf{x}_t \oplus \mathbf{c}$, where the image $\mathbf{c}$ serves as a condition. Then, we directly feed $\mathbf{a}_t$ into the DiT. The first layer of DiT is a patchify operation, which converts the spatial input $\mathbf{a}_t$ into a 1D sequence of $T$ tokens (patches), each with a dimension of $D$, by linearly embedding each patch of size $p \times p$ from the input $\mathbf{a}_t$. Subsequently, the input tokens are processed by a sequence of Transformer blocks, called DiT blocks. After the final DiT block, each token is linearly projected into a $p \times p$ tensor, which is then reshaped back to the original spatial resolution to obtain the predicted velocity $\mathbf{v}_t$ (*i.e.*, $\mathbf{x}_1 - \mathbf{x}_0$), with a channel dimension of 1.

Unfortunately, performing diffusion directly in the pixel space leads to poor convergence and highly inaccurate depth predictions. As shown in Figure 6, the model struggles to model both global image structure and fine-grained details. To address this, we extract high-level semantic representations $\mathbf{e}$ as guidance from the input image $\mathbf{c}$ using a vision foundation model $f$, as follows:

$$\mathbf{e} = f(\mathbf{c}) \in \mathbb{R}^{T' \times D'}, \tag{5}$$

where $T'$ and $D'$ are the number of tokens and the embedding dimension of $f(\mathbf{c})$, respectively. These high-level semantic representations are then incorporated into our DiT model, enabling it to more effectively preserve global semantic consistency while enhancing fine-grained visual details. However, we found that the magnitude of the obtained semantics $\mathbf{e}$ differs significantly from the magnitude of the tokens in our DiT model, which affects both the stability of the model's training and its performance. To address this, we normalize the semantic representation $\mathbf{e}$ along the feature dimension using L2 norm, as follows:

$$\hat{\mathbf{e}} = \frac{\mathbf{e}}{\|\mathbf{e}\|_2}. \tag{6}$$

Subsequently, the normalized semantic representation is integrated into the tokens $\mathbf{z}$ of our DiT model via a multilayer perceptron (MLP) layer $h_\phi$,

$$\mathbf{z}' = h_\phi(\mathbf{z} \oplus \mathcal{B}(\hat{\mathbf{e}})), \tag{7}$$

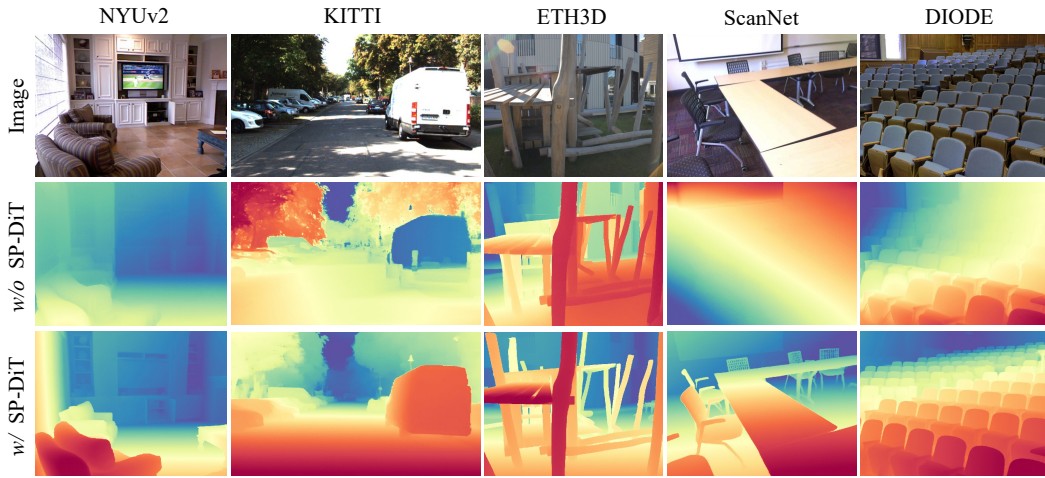

Figure 6: **Qualitative ablations for the proposed SP-DiT.** Without SP-DiT, the vanilla DiT model struggles with preserving global semantics and generating fine-grained visual details.

where $\mathcal{B}(\cdot)$ denotes the bilinear interpolation operator, which aligns the spatial resolution of the semantic representation $\hat{e}$ with that of the DiT tokens. The resulting $z'$ denotes the DiT tokens enhanced with semantics. After the fusion, the subsequent DiT blocks are prompted by semantics to effectively preserve global semantic consistency while enhancing fine-grained visual details in the high-resolution pixel space. We refer to these subsequent DiT blocks as Semantics-Prompted DiT.

In this work, we experiment with various pretrained vision foundation models, including DINOv2 [44], VGGT [65], MAE [24], and Depth Anything v2 [82]. All of them significantly boost performance and facilitate more stable and efficient training, as shown in Table 3. Note that we only utilize the encoder of each vision foundation model, *e.g.*, a 24-layer Vision Transformer encoder (ViT-L/14) for both DINOv2 [44] and Depth Anything v2 [82].

### 3.4 Cascade DiT Design

Although the proposed Semantics-Prompted DiT significantly improves accuracy performance, performing diffusion directly in the pixel space remains computationally expensive. To address this issue, We propose a novel Cascaded DiT Design to reduce the computational burden of the model. We observe that in DiT architectures, the early blocks are primarily responsible for capturing global image structures and low-frequency information, while the later blocks focus on modeling fine-grained, high-frequency details.

To optimize the efficiency and effectiveness of this process, we adopt a large patch size in the early DiT blocks. This design significantly reduces the number of tokens that need to be processed, leading to lower computational cost. Additionally, it encourages the model to prioritize learning and modeling global image structures and low-frequency information, which also better aligns with the high-level semantic representations extracted from the input image. In the later DiT blocks, we increase the number of tokens, which is equivalent to using a smaller patch size. This allows the model to better focus on fine-grained spatial details. The resulting coarse-to-fine cascaded design mirrors the hierarchical nature of visual perception and improves both the efficiency and accuracy of depth estimation.

Specifically, for our diffusion model with a total of $N$ DiT blocks, the first $N/2$ blocks constitute the coarse stage with a larger patch size, while the remaining $N/2$ blocks (*i.e.*, SP-DiT) form the fine stage using a smaller patch size.

### 3.5 Implementation Details

In this section, we provide essential information about the model architecture details, depth normalization, and training details.

Table 1: **Zero-shot relative depth estimation.** Better: AbsRel $\downarrow$, $\delta_1$ $\uparrow$. **Bold** numbers are the best. Our model outperforms other generative models on five benchmarks. Ours (512) represents $512 \times 512$ model, and Ours (1024) represents $1024 \times 768$ model.

| Type | Method | Training Data | NYUv2 | | KITTI | | ETH3D | | ScanNet | | DIODE | |
|---|---|---|---|---|---|---|---|---|---|---|---|---|
| | | | AbsRel$\downarrow$ | $\delta_1\uparrow$ | AbsRel$\downarrow$ | $\delta_1\uparrow$ | AbsRel$\downarrow$ | $\delta_1\uparrow$ | AbsRel$\downarrow$ | $\delta_1\uparrow$ | AbsRel$\downarrow$ | $\delta_1\uparrow$ |
| *Discriminative* | DiverseDepth[87] | 320K | 11.7 | 87.5 | 19.0 | 70.4 | 22.8 | 69.4 | 10.9 | 88.2 | - | - |
| | MiDaS[49] | 2M | 11.1 | 88.5 | 23.6 | 63.0 | 18.4 | 75.2 | 12.1 | 84.6 | - | - |
| | LeReS[89] | 354K | 9.0 | 91.6 | 14.9 | 78.4 | 17.1 | 77.7 | 9.1 | 91.7 | - | - |
| | Omnidata[13] | 12M | 7.4 | 94.5 | 14.9 | 83.5 | 16.6 | 77.8 | 7.5 | 93.6 | - | - |
| | HDN[91] | 300K | 6.9 | 94.8 | 11.5 | 86.7 | 12.1 | 83.3 | 8.0 | 93.9 | - | - |
| | DPT[48] | 1.2M | 9.8 | 90.3 | 10.0 | 90.1 | **7.8** | **94.6** | 8.2 | 93.4 | - | - |
| | DepthAny. v2[82] | 54K | 5.4 | 97.2 | 8.6 | 92.8 | 12.3 | 88.4 | - | - | 8.8 | 93.7 |
| | DepthAny. v2[82] | 62M | **4.5** | **97.9** | **7.4** | **94.6** | 13.1 | 86.5 | **6.5** | **97.2** | 6.6 | 95.2 |
| *Generative* | Marigold[34] | 74K | 5.5 | 96.4 | 9.9 | 91.6 | 6.5 | 96.0 | 6.4 | 95.1 | 10.0 | 90.7 |
| | GeoWizard[17] | 280K | 5.2 | 96.6 | 9.7 | 92.1 | 6.4 | 96.1 | 6.1 | 95.3 | 12.0 | 89.8 |
| | DepthFM[20] | 74K | 5.5 | 96.3 | 8.9 | 91.3 | 5.8 | 96.2 | 6.3 | 95.4 | - | - |
| | GenPercept[78] | 90K | 5.2 | 96.6 | 9.4 | 92.3 | 6.6 | 95.7 | 5.6 | 96.5 | - | - |
| | Lotus[23] | 54K | 5.4 | 96.8 | 8.5 | 92.2 | 5.9 | 97.0 | 5.9 | 95.7 | 9.8 | 92.4 |
| | Ours (512) | 54K | 4.3 | 97.4 | 8.0 | 93.1 | 4.5 | 97.7 | **4.5** | **97.3** | 7.0 | 95.5 |
| | Ours (1024) | 125K | **4.1** | **97.7** | **7.0** | **95.5** | **4.3** | **98.0** | 4.6 | 97.2 | **6.8** | **95.9** |

**Model architecture details.** In our implementation, we use a total of $N = 24$ DiT blocks, each operating at a hidden dimension of $D = 1024$. The first 12 blocks are standard DiT blocks with a patch size of 16, corresponding to $(H/16) \times (W/16)$ tokens for an input of size $H \times W$. After the 12th block, we employ an MLP layer to expand the hidden dimension by a factor of 4, followed by reshaping to obtain $(H/8) \times (W/8)$ tokens. The remaining 12 SP-DiT blocks then further process these $(H/8) \times (W/8)$ tokens. Finally, we employ an MLP layer followed by a reshaping operation to transform the processed tokens into an $H \times W$ depth map. In contrast to prior monocular depth models, such as Depth Anything and Depth Pro, our model does not rely on any convolutional layers.

**Depth normalization.** The ground truth depth values are normalized to match the scale expected by the diffusion model. Before normalization, we convert the depth values into log scale to ensure a more balanced capacity allocation across both indoor and outdoor scenes. Specifically, we apply the transformation $\tilde{\mathbf{d}} = \log(\mathbf{d} + \epsilon)$, where $\tilde{\mathbf{d}}$ denotes the transformed depth, $\mathbf{d}$ is the original depth value, and $\epsilon$ is a small positive constant (*e.g.*, 1) to ensure numerical stability. We then normalize the log-scaled depth $\tilde{\mathbf{d}}$ using:

$$\hat{\mathbf{d}} = \frac{\tilde{\mathbf{d}} - d_{\min}}{d_{\max} - d_{\min}} - 0.5, \tag{8}$$

where $d_{min}$ and $d_{max}$ are the 2% and 98% depth percentiles of each map, respectively.

**Training details.** We train two variants of the diffusion model at different resolutions: one at $512 \times 512$ and the other at $1024 \times 768$. We train all models on 8 NVIDIA GPUs with a per-GPU batch size of 4, using the AdamW optimizer with a constant learning rate of $1 \times 10^{-4}$. The training loss is the MSE loss between the predicted and true velocity, as shown in Equation 3, and the gradient matching loss, which is adopted from [82].

## 4 Experiments

### 4.1 Experimental Setup

**Training datasets.** Our objective is to estimate pixel-perfect depth maps, which, when converted to point clouds, are free of *flying pixels* and geometric artifacts. To achieve this, it is essential to train on datasets with high-quality ground truth point clouds. We adopt Hypersim [50], a photorealistic synthetic dataset with accurate and clean 3D geometry, which contains approximately 54K samples, to train the $512 \times 512$ model. For the $1024 \times 768$ model, we additionally leverage four datasets, UrbanSyn [19] (7.5K), UnrealStereo4K [62] (8K), VKITTI [5] (25K), and TartanAir [71] (30K), to further enhance the model's generalization and robustness.

Table 2: **Ablation studies on five zero-shot benchmarks.** All metrics are presented in percentage terms, **bold** numbers are the best. Inference time was tested on an RTX 4090 GPU. All results were obtained using the $512 \times 512$ model.

| Method | NYUv2 | | KITTI | | ETH3D | | ScanNet | | DIODE | | Time(s) |
|---|---|---|---|---|---|---|---|---|---|---|---|
| | AbsRel↓ | $\delta_1$↑ | AbsRel↓ | $\delta_1$↑ | AbsRel↓ | $\delta_1$↑ | AbsRel↓ | $\delta_1$↑ | AbsRel↓ | $\delta_1$↑ | |
| DiT (vanilla) | 22.5 | 72.8 | 27.3 | 63.9 | 12.1 | 87.4 | 25.7 | 65.1 | 23.9 | 76.5 | 0.19 |
| SP-DiT | 4.8 | 96.7 | 8.6 | 92.2 | 4.6 | 97.5 | 6.2 | 94.8 | 8.2 | 94.1 | 0.20 |
| SP-DiT+Cas-DiT | **4.3** | **97.4** | **8.0** | **93.1** | **4.5** | **97.7** | **4.5** | **97.3** | **7.0** | **95.5** | **0.14** |

**Evaluation setup.** Following the majority of previous depth estimation models [34, 17, 23], we evaluate the zero-shot relative depth estimation performance on five real-world datasets: NYUv2 [58], KITTI [18], ETH3D [56], ScanNet [11], and DIODE [63], covering both indoor and outdoor scenes. To assess the quality of depth estimation, we adopt two widely-used evaluation metrics: Absolute Relative Error (AbsRel) and $\delta_1$ accuracy. To demonstrate that our model generates point clouds without *flying pixels*, we convert the estimated depth maps into 3D point clouds and evaluate them using the proposed edge-aware metric. For simplicity, the majority of quantitative evaluations are conducted using the $512 \times 512$ model. We employ the $1024 \times 768$ model for the quantitative evaluations in Table 1 as well as for qualitative comparisons.

## 4.2 Ablations and Analysis

**Component-wise ablation analysis.** We adopt the vanilla DiT [45] model as our baseline and conduct ablations on our proposed modules. Quantitative results are shown in Table 2. Directly performing diffusion generation in high-resolution pixel space is highly challenging due to substantial computational costs and optimization difficulties, leading to significant performance degradation. As illustrated in Figure 6, the baseline model struggles with preserving global semantics and generating fine-grained visual details. In contrast, the proposed Semantics-Prompted DiT (SP-DiT) addresses these challenges, achieving significantly improved accuracy, for example, a 78% gain on the NYUv2 AbsRel metric. We further introduce a novel Cascaded DiT Design (Cas-DiT) that progressively increases the number of tokens. This coarse-to-fine design not only significantly improves efficiency, for example, reducing inference time by 30% on an RTX 4090 GPU, but also better models global context, leading to noticeable gains in accuracy.

**Ablations on vision foundation models (VFMs).** We evaluate the performance of SP-DiT using pretrained vision encoders from different VFMs, including MAE [24], DINOv2 [44], Depth Anything v2 [82], and VGGT [65], as illustrated in Table 3. All of them significantly boost performance.

## 4.3 Zero-Shot Relative Depth Estimation

To evaluate our model's zero-shot generalization, we compare it with recent depth estimation models [82, 4, 34, 23, 20] on five real-world benchmarks. As shown in Table 1, our model outperforms all other generative depth estimation models for all evaluation metrics. Unlike previous generative models, we do not rely on image priors from a pretrained Stable Diffusion [51] model. Instead, our diffusion model is trained from scratch and still achieves superior performance. Our model generalizes well to a wide range of real-world scenes, even when trained solely on synthetic depth datasets. Visual comparisons are shown in Figure 4, our model (1024) preserves more fine-grained details than Depth Anything v2 [82] and MoGe 2 [70]. Moreover, it demonstrates significantly higher robustness than Depth Pro [4], especially in challenging regions with complex textures, cluttered backgrounds, or large sky areas.

## 4.4 Edge-Aware Point Cloud Evaluation

Our objective is to estimate pixel-perfect depth maps that yield clean point clouds without *flying pixels*, which often occur at object edges due to inaccurate depth predictions in these regions. However, existing evaluation benchmarks and metrics often struggle to reflect *flying pixels* at object edges. For

Table 3: **Ablation studies on Vision Foundation Models (VFMs).** Note that we only utilize a pretrained encoder from these VFMs, such as a 24-layer ViT from DINOv2 or Depth Anything v2.

| VFM Type | NYUv2 | | KITTI | | ETH3D | | ScanNet | | DIODE | |
|---|---|---|---|---|---|---|---|---|---|---|
| | AbsRel↓ | $\delta_1$↑ | AbsRel↓ | $\delta_1$↑ | AbsRel↓ | $\delta_1$↑ | AbsRel↓ | $\delta_1$↑ | AbsRel↓ | $\delta_1$↑ |
| w/o SP-DiT | 22.5 | 72.8 | 27.3 | 63.9 | 12.1 | 87.4 | 25.7 | 65.1 | 23.9 | 76.5 |
| SP-DiT (MAE [24]) | 6.4 | 95.0 | 14.4 | 84.9 | 7.3 | 94.8 | 7.7 | 92.5 | 11.6 | 91.3 |
| SP-DiT (DINOv2 [44]) | 4.8 | 96.4 | 9.3 | 91.2 | 5.6 | 96.2 | 5.1 | 96.9 | 9.2 | 93.5 |
| SP-DiT (VGGT [65]) | 4.7 | 96.7 | **7.6** | **94.1** | **4.1** | **97.8** | **3.8** | **98.0** | 7.8 | 94.9 |
| SP-DiT (DepthAny. v2 [82]) | **4.3** | **97.4** | 8.0 | 93.1 | 4.5 | 97.7 | 4.5 | 97.3 | **7.0** | **95.5** |

Table 4: **Edge-aware point cloud evaluation.** Our model achieves the best performance on the high-quality Hypersim test set. To further verify that VAE compression leads to *flying pixels*, we evaluate the ground truth depth maps after VAE reconstruction, denoted as GT(VAE).

| | Marigold[34] | GeoWizard[17] | DepthAny. v2[82] | DepthPro[4] | GT(VAE) | Ours |
|---|---|---|---|---|---|---|
| Chamfer Dist.↓ | 0.17 | 0.16 | 0.18 | 0.14 | 0.12 | **0.08** |

example, benchmarks like NYUv2 or KITTI usually lack edge annotations, while metrics such as AbsRel and $\delta_1$ are dominated by flat regions, making it difficult to assess depth accuracy at edges.

To address these limitations, we evaluate on the official test split of the Hypersim [50] dataset, which provides high-quality ground-truth point clouds and is not used during training. We further propose an edge-aware point cloud metric that quantifies depth accuracy at edges. Specifically, we extract edge masks from ground-truth depth maps using the Canny operator and compute the Chamfer Distance between predicted and ground-truth point clouds near these edges.

Quantitative results in Table 4 show that our method achieves the best performance. Discriminative models like Depth Pro [4] and Depth Anything v2 [82] tend to smooth edges, causing *flying pixels*. Generative models such as Marigold [34] rely on VAE compression, which blurs edges and details, causing artifacts in the reconstructed point clouds. To illustrate this, we encode and decode the ground-truth depth using a VAE (GT(VAE)), without any generative process. Table 4 and Figure 2 show that VAE compression introduces *flying pixels*, leading to a larger Chamfer Distance than ours.

## 5 Conclusion

We presented **Pixel-Perfect Depth**, a monocular depth estimation model that leverages pixel-space diffusion transformers to produce high-quality, flying-pixel-free point clouds. Unlike prior generative depth models that rely on latent-space diffusion with a VAE, our model performs diffusion directly in the pixel space, avoiding *flying pixels* caused by VAE compression. To tackle the complexity and optimization challenges of pixel-space diffusion, we introduce Semantics-Prompted DiT and Cascade DiT Design, which greatly boost performance. Our model significantly outperforms prior models in edge-aware point cloud evaluation.

**Limitations and future work.** This work has two known limitations. First, like most image-based diffusion models, it lacks temporal consistency when applied to video frames, resulting in a little flickering depth across frames. Second, its multi-step diffusion process leads to slower inference compared to discriminative models like Depth Anything v2. Future works can address these limitations by exploring video depth estimation methods [57, 31, 80, 6, 33] to improve temporal consistency and adopting DiT acceleration strategies to speed up inference.

**Acknowledgements.** This research is supported by the National Key R&D Program of China (2024YFE0217700), the National Natural Science Foundation of China (623B2036, 62472184), the Fundamental Research Funds for the Central Universities, and the Innovation Project of Optics Valley Laboratory (Grant No. OVL2025YZ005).

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

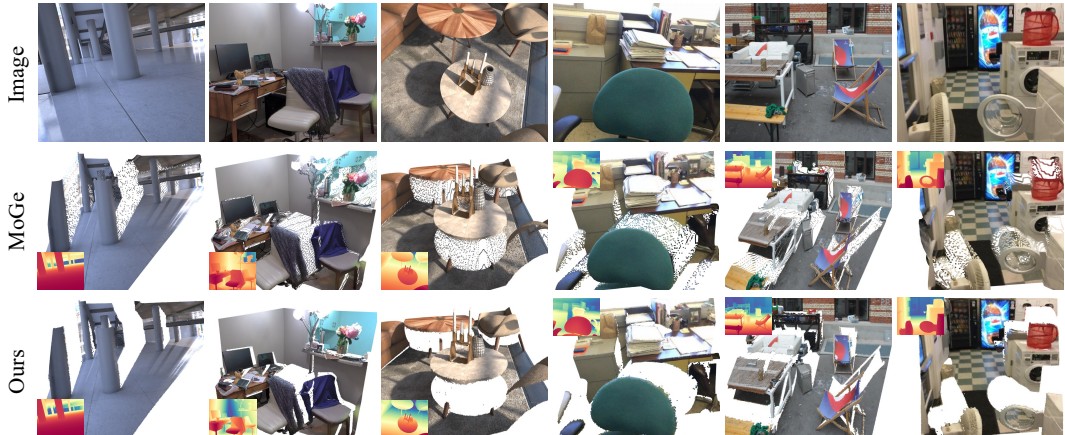

Figure 7: **Qualitative comparisons with MoGe [69].** Top: input images are taken from four test sets: Hypersim [50], DIODE [63], ScanNet [11], and ETH3D [56]. Middle: results of MoGe [69]. Bottom: our results. As a discriminative model, MoGe [69], like other discriminative models [82, 4], also suffers from *flying pixels* at edges and details.

Table 5: **Quantitative comparisons with REPA [90].** Our model significantly outperforms REPA [90]. To ensure a fair comparison, the pretrained vision encoder used in both DiT+REPA and DiT+Ours is kept the same.

| Method | NYUv2 | | KITTI | | ETH3D | | ScanNet | | DIODE | |
|---|---|---|---|---|---|---|---|---|---|---|
| | AbsRel↓ | $\delta_1$↑ | AbsRel↓ | $\delta_1$↑ | AbsRel↓ | $\delta_1$↑ | AbsRel↓ | $\delta_1$↑ | AbsRel↓ | $\delta_1$↑ |
| DiT (vanilla) | 22.5 | 72.8 | 27.3 | 63.9 | 12.1 | 87.4 | 25.7 | 65.1 | 23.9 | 76.5 |
| DiT+REPA [90] | 17.6 | 78.0 | 23.4 | 70.6 | 9.1 | 91.2 | 20.1 | 74.3 | 14.6 | 86.9 |
| DiT+Ours | **4.3** | **97.4** | **8.0** | **93.1** | **4.5** | **97.7** | **4.5** | **97.3** | **7.0** | **95.5** |

## A Qualitative Comparisons with MoGe

We provide qualitative comparisons of reconstructed point clouds, as shown in Figure 7. MoGe [69], as a discriminative model, suffers from *flying pixels* at edges and fine structures, a common issue observed in other discriminative models [82, 4]. Our model produces significantly fewer *flying pixels* compared to MoGe [69].

## B Additional Discussion with REPA

We provide an additional discussion on the recent image generation method REPA [90]. REPA [90] aligns intermediate tokens in diffusion models with pretrained vision encoder, significantly improving training efficiency and generation quality for image generation tasks. We compare our method with REPA [90], and the quantitative evaluation results are presented in Table 5. DiT+REPA refers to training the DiT model with REPA's representation alignment, while DiT+Ours denotes training the DiT model using our Semantics-Prompted DiT. For a fair comparison, the pretrained vision encoder used in both DiT+REPA and DiT+Ours is kept the same. Experimental results show that our Semantics-Prompted DiT significantly outperforms REPA [90]. We attribute our model's superiority over REPA to two factors. First, during training, REPA's implicit alignment of DiT tokens with the pretrained vision encoder is suboptimal, making it difficult for DiT to effectively leverage semantic prompts from the pretrained vision encoder. In contrast, our Semantics-Prompted DiT directly integrates semantic cues, resulting in more effective prompts. Second, at inference, REPA cannot leverage the pretrained vision encoder to provide semantic prompts, whereas our method effectively incorporates high-level semantics into the Semantics-Prompted DiT during inference to prompt the diffusion process.

Table 6: **Runtime comparison on RTX 4090 GPU.** The runtime is measured using the $512 \times 512$ model with 4 denoising steps.

|  | Depth Anything v2 [82] | DepthPro [4] | PPD-Large | PPD-Small |
|---|---|---|---|---|
| Time (ms) | 18 | 170 | 140 | 40 |

Table 7: **Quantitative comparisons between PPD-Large and PPD-Small.**

| Method | NYUv2 | | KITTI | | ETH3D | | ScanNet | | DIODE | |
|---|---|---|---|---|---|---|---|---|---|---|
| | AbsRel↓ | $\delta_1$↑ | AbsRel↓ | $\delta_1$↑ | AbsRel↓ | $\delta_1$↑ | AbsRel↓ | $\delta_1$↑ | AbsRel↓ | $\delta_1$↑ |
| PPD-Small | 4.5 | 97.3 | 8.3 | 92.8 | 4.6 | 97.4 | 4.7 | 97.2 | 7.3 | 95.3 |
| PPD-Large | **4.3** | **97.4** | **8.0** | **93.1** | **4.5** | **97.7** | **4.5** | **97.3** | **7.0** | **95.5** |

## C    Analysis of Flying Pixels in Different Types of VAEs

To better understand the emergence of *flying pixels* in VAE-based reconstructions, we analyze VAEs with different latent dimensions (*i.e.*, channel) by using them to reconstruct ground truth depth maps. Figure 8 shows that both VAE variants exhibit *flying pixels* at object edges and details, revealing a common weakness of VAE reconstructions in preserving precise geometric structures. VAE-d4 (SD2) denotes the reconstruction of ground truth depth maps using the VAE from Stable Diffusion 2, with a latent dimension of 4, which is also used in Marigold [34]. VAE-d16 (SD3.5) uses the VAE from Stable Diffusion 3.5, which has a latent dimension of 16.

## D    Efficiency and Lightweight Variant

Our Pixel-Perfect Depth (PPD) model is slower than Depth Anything v2 [82] owing to the multi-step diffusion process, but its inference time remains comparable to Depth Pro [4], as shown in Table 6. To further accelerate inference, we develop a lightweight variant, PPD-Small, which achieves substantially faster runtime with only marginal accuracy loss, as shown in Table 7. In contrast to PPD-Large, PPD-Small is built upon DiT-Small with a reduced number of parameters, making it more suitable for efficient inference.

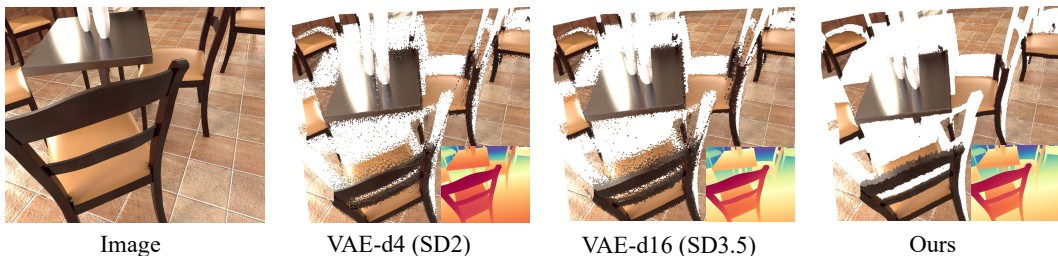

| Image | VAE-d4 (SD2) | VAE-d16 (SD3.5) | Ours |
|---|---|---|---|

Figure 8: **Validation of flying pixels in different types of VAEs.** We present further qualitative comparisons showing that increasing the latent dimension in VAEs fails to eliminate *flying pixels*. VAE-d4 (SD2) denotes the reconstruction of ground truth depth maps using the VAE from Stable Diffusion 2, with a latent dimension of 4, which is also used in Marigold. VAE-d16 (SD3.5) uses the VAE from Stable Diffusion 3.5, which has a latent dimension of 16.

