# OpenReview forum: "Pixel-Perfect Depth with Semantics-Prompted Diffusion Transformers"
_NeurIPS.cc/2025/Conference — NeurIPS 2025 poster_

### Official Review · Reviewer_icJf · 2025-06-07

**Clarity:** 4
**Significance:** 3
**Originality:** 4
**Rating:** 5
**Confidence:** 5

**Summary:**

The authors present an innovative approach to relative depth estimation using Semantics-Guided Diffusion Transformers (SG-DiT). Focused on generating high-quality depth maps, the paper effectively addresses challenges such as flying pixels and edge artifacts. The claims are robustly supported by both theoretical foundations and experimental results throughout the study, showcasing the model's ability to generalize across diverse scenarios. Furthermore, the paper provides insights into the computational efficiency of the proposed methods. Overall, it is well-structured, thoroughly addressing vital issues concerning generalization and reproducibility, thereby marking a significant advancement in the domain of more precise relative depth estimation.

**Questions:**

See in the weaknesses part.

**Ethical Concerns:**

["NO or VERY MINOR ethics concerns only"]

**Final Justification:**

The author's response has addressed my concerns, so I have decided to accept the manuscript.

**Limitations:**

yes

**Paper Formatting Concerns:**

Not noticed.

**Quality:**

4

**Strengths And Weaknesses:**

Strengths: The paper presents detailed theoretical and experimental validation, thoroughly discussing the model's architecture while providing quantitative evaluation metrics and visual quality demonstrations. It proposes an innovative solution for monocular relative depth estimation, with considerable potential for practical applications. The paper effectively addresses typical issues related to flying pixels and edge artifacts, offering robust comparisons and solutions.

Weaknesses:

- It seems that post-processing techniques, such as filtering depth at certain positions using image gradient information, could also resolve the flying pixels issue?

- Compared to relative depth, monocular metric depth estimation is more practical. Can the method proposed in this paper be extended to monocular metric depth estimation?

- How does this method perform in extremely detailed regions of an image, such as hair details showcased in Depth Pro?

---

> ### Author Rebuttal · Authors · 2025-07-28
>
> Thank you for your positive feedback and recognition of our **detailed theoretical and experimental validation**, **innovative approach**, and **practical potential**. We've addressed all your comments in detail below.
>
> ### **Q1: Filtering depth.**
>
> Thank you for the thoughtful question. While post-processing techniques, such as filtering depth based on image gradients, may offer some mitigation, they are often ineffective in handling severe flying-pixel artifacts. Moreover, such techniques typically smooth or replace flying pixels using neighboring values, which can result in edge blurring or structural distortion.
>
> In contrast, our model addresses flying pixels directly during the prediction stage, enabling it to preserve sharp object boundaries and produce structurally accurate and reliable depth maps, without relying on heuristic post-hoc fixes.
>
> ### **Q2: Monocular metric depth estimation.**
>
> Our method can be extended to metric depth estimation using metric depth normalizing techniques proposed in Metric3D [Yin et al., ICCV 2023] and Depth Pro [Bochkovskii el al., ICLR 2025]. Specifically, our method could serve as backbone network to predict metric depth normalized by focal length, obtaining metric depth without flying pixels. This could be one of the interesting future works.
>
> ### **Q3: Performance in extremely detailed regions.**
>
> Our method performs well in fine-detail regions such as hair structures, which is one of the core strengths of diffusion-based generative models, as they are particularly effective at modeling high-frequency details. We have visualized hair details similar to those in Depth Pro. Due to the limitations of the rebuttal format, we are unable to include these qualitative visualizations here, but we will ensure that they are incorporated into the final version of the paper.

---

> > ### Comment · Reviewer_icJf · 2025-08-02
> >
> > Thank you for the response. The authors have promised certain improvements and have addressed my main concerns. Given the current state of the submission, I am inclined to maintain my score (or to increase it from 4 to 4.5).

---

> > > ### Author Response · Authors · 2025-08-05
> > >
> > > Thank you for your response and for considering a higher score. Once again, we sincerely appreciate your thorough review and insightful feedback.

---

### Official Review · Reviewer_fjSa · 2025-06-22

**Clarity:** 4
**Significance:** 4
**Originality:** 4
**Rating:** 5
**Confidence:** 4

**Summary:**

This paper introduces Pixel-Perfect Depth, a monocular depth estimation framework that operates in pixel space rather than the more common latent space, thereby avoiding flying-pixel artifacts introduced by VAE compression. To address the complexity of high-resolution generation, the authors propose two novel architectural components: Semantics-Guided Diffusion Transformers (SG-DiT) and a Cascade DiT design (Cas-DiT). This method achieves SOTA results and introduces an edge-aware evaluation metric for assessing depth quality near object boundaries. I find the results promising, and I believe the paper would be further strengthened if the experimental issues noted in the Weaknesses section are addressed.

**Questions:**

Please refer to the Weaknesses. If the authors can adequately address those concerns in the rebuttal, I would consider increasing my score from 4 to 5.

**Ethical Concerns:**

["NO or VERY MINOR ethics concerns only"]

**Final Justification:**

I have read the rebuttal and appreciate the authors’ clarifications. They have addressed my main concerns, and I am increasing my score from 4 to 5.

**Limitations:**

yes

**Quality:**

4

**Strengths And Weaknesses:**

Strengths:
1. The motivation is clear and well-justified: the method targets the flying pixel issue in depth estimation models, particularly at object boundaries. By proposing a pixel-space diffusion framework with semantic guidance, the authors offer a novel and effective solution. The qualitative and quantitative results across diverse benchmarks demonstrate strong performance.
2. The paper introduces a new edge-aware evaluation metric designed to quantify depth estimation accuracy specifically at object boundaries. This addresses a gap in current evaluation protocols and is a meaningful contribution to the field.

Weaknesses:
1. In the qualitative ablation analysis (Figure 5), only depth map visualizations are shown. Including corresponding point cloud visualizations (as in Figure 1) would help better assess how this component impacts the flying-pixel artifacts.
2. Since the core motivation of the paper is to reduce flying pixels, and a new edge-aware evaluation metric is proposed for that purpose, it would strengthen the paper to conduct ablation studies using this metric. This would provide more direct evidence of how each module contributes to mitigating boundary errors.

---

> ### Author Rebuttal · Authors · 2025-07-28
>
> Thank you for your very positive feedback! We truly appreciate you highlighting the **well-motivated** design of our method, its **novel solution** to the flying pixel issue, and the **significant contribution** of our new edge-aware evaluation metric. We've addressed all your comments in detail below.
>
> ### **Q1: Point cloud visualizations for ablation analysis.**
>
> Thank you for the valuable suggestion. We have visualized the corresponding point clouds for the ablation studies  (with and without SG-DiT) to assess the impact on flying-pixel artifacts. Our SG-DiT not only effectively reduces flying pixels but also better preserves the global structure of the scene. Due to space limitations in the rebuttal, we are unable to include these figures here, but we will ensure that these visualizations are included in the final version of the paper.
>
> ### **Q2: Ablation studies on edge-aware evaluation metric.**
>
> Thank you for the insightful suggestion. We have already included ablation studies using the proposed edge-aware evaluation metric, as shown in the table below.
>
> |        Method      |   Chamfer Distance ↓   |
> |--------|:--------:|
> |  DiT (baseline)  |  0.13  |
> |  SG-DiT  |  0.09  |
> |  SG-DiT+Cas-DiT (full)  |  0.08  |
>
> Experimental results demonstrate that our proposed components consistently reduce flying pixels and enhance the structural integrity of the reconstructed point clouds. Specifically, SG-DiT improves the Chamfer Distance from 0.13 to 0.09, while the full model further improves it to 0.08, confirming the effectiveness of our design in mitigating boundary errors.

---

> > ### Comment · Reviewer_fjSa · 2025-08-01
> >
> > Thank you for the response. I have read the rebuttal and the other reviews. As the authors have addressed my main concerns, I am inclined to revise my score from 4 to 5 based on the current state of the submission.

---

> > > ### Author Response · Authors · 2025-08-01
> > >
> > > Thank you for acknowledging our response and for your willingness to raise the score to 5. We truly appreciate your constructive feedback. Thank you again for your thoughtful review!

---

### Official Review · Reviewer_MxaR · 2025-06-29

**Clarity:** 3
**Significance:** 3
**Originality:** 2
**Rating:** 5
**Confidence:** 3

**Summary:**

This paper analyzes the limitations of current generative and discriminative models for monocular depth estimation. Specifically, it identifies two major issues: (1) the training instability and poor performance of pixel-based diffusion models, and (2) the boundary over-smoothing problem in discriminative models. To address these challenges, the authors propose a novel geometry-guided function model, which effectively stabilizes the training of diffusion models and preserves sharp depth boundaries. The experimental results support the authors' analysis and demonstrate that their method achieves state-of-the-art performance with a relatively simple network architecture.

**Questions:**

1. What is the inference time for a 512x512 images.
2.The paper mentions that the geometry features are normalized due to their smaller magnitude compared to diffusion features. However, if the diffusion features are not globally normalized either, wouldn't the scale mismatch still affect feature fusion? Would it be more appropriate to normalize both or apply a learnable scaling? Would you mind do an ablation about this?

**Ethical Concerns:**

["NO or VERY MINOR ethics concerns only"]

**Final Justification:**

The authors have addressed my problem. It is good work.

**Limitations:**

Yes

**Quality:**

3

**Strengths And Weaknesses:**

Strengths:
1. To the best of my knowledge, this is the first work that combines a foundation geometry model with a pixel-space diffusion model for monocular depth estimation. Pixel-based diffusion model is particularly meaningful, as latent-space representations inevitably suffer from information loss. Therefore, the proposed approach is both novel and potentially impactful.
2. The paper provides insightful analysis of the limitations in current depth estimation models, which may inspire future research directions. Moreover, the proposed solution is simple yet effective. By adopting a pure-transformer architecture, the method remains lightweight and easily extensible, which is beneficial for follow-up work.
3.The experiments are comprehensive and convincingly demonstrate that the proposed method achieves state-of-the-art performance.
4. The writing is clear, well-organized, and easy to follow.

WeakNesses.

1.Although the simple geometry integration achieves SOTA, it is unclear if more efficient or better alternatives than normalize + concat exist. The improvement over DepthAnything v2 is not very significant, and more ablation studies are needed to justify this design choice.

2. The paper lacks a thorough analysis of efficiency. While it mentions that diffusion models inevitably slow down inference, it does not quantify how much slower they are. A more precise comparison of inference time would be necessary.

3. The paper lacks deeper theoretical insight. While it is somewhat expected that geometry priors can enhance diffusion-based depth estimation, the paper does not clearly analyze the underlying challenges or implications of combining the two. As a result, the novelty seems to reside more in the practical combination of known components than in a fundamentally new conceptual contribution.

---

> ### Author Rebuttal · Authors · 2025-07-28
>
> Thank you for your positive feedback and recognition of our work’s **novelty**, **simplicity**, and **effectiveness**. We address your questions as follows.
>
> ### **Q1: Simple geometry integration achieves SOTA, it is unclear if more efficient or better alternatives than normalize + concat exist.**
>
> We observed that geometry features (semantic features) have much larger magnitudes than diffusion features, which caused training instability and frequent NaNs when fused directly. To address this, we normalize the geometry features to match the scale of diffusion features. This simple strategy stabilizes training and achieves strong performance. We also experimented with more complex alternatives, such as learnable scaling or adaptive fusion weights, but they did not bring clear performance gains.
>
> ### **Q2: Improvement over Depth Anything v2.**
>
> The main goal of our work is to address the flying pixels problem commonly observed in discriminative models such as Depth Anything v2. As shown in Figures 1 and 4 and Table 4 of our paper, our model significantly reduces flying pixels. While standard benchmarks and metrics (e.g., Table 1 in our paper) typically fail to assess flying pixels at object edges, we propose an edge-aware point cloud metric to quantify flying pixels specifically at edges. This is explained in Section 4.4 of the paper. As illustrated in Table 4 of our paper, our method achieves a Chamfer distance of 0.08, which is a substantial improvement compared to 0.18 from Depth Anything v2.
>
> ### **Q3: Analysis of efficiency.**
>
> We provide a quantitative comparison of inference speed against other representative models. For a fair comparison, all models are evaluated on an RTX 4090 GPU with an input resolution of 512×512. As shown in the table below, our model is slower than Depth Anything v2, primarily due to the multi-step nature of diffusion-based inference. To offer a clearer view of the computational overhead, we report the runtime of each component: the semantics encoder takes 10 ms, and each diffusion iteration takes approximately 32.5 ms. With 4 iterations, the total inference time sums to 140 ms.
>
> |      Method      |   Time (ms)   |
> |--------|:--------:|
> |    Depth Anything v2   |    18    |
> |    Depth Pro   |    170    |
> |    Marigold   |    210    |
> |    Ours   |    140    |
>
>
>
> |      Module      |   Time (ms)   |
> |--------|:--------:|
> |    Semantics encoder   |    10    |
> |    1 iter (diffusion)   |    32.5    |
> |    4 iter (diffusion)    |    130    |
> |    Ours (total)   |    140    |
>
> ### **Q4: Deeper theoretical insight.**
>
> Our work is grounded in a principled analysis of the flying pixels problem commonly found in discriminative models such as Depth Anything v2 and Depth Pro, which tend to predict averaged depth values at depth-discontinuous edges due to the objective of minimizing regression loss. In contrast, our pixel-space diffusion model estimates pixel-wise depth distributions, allowing it to preserve sharper edges and avoid flying pixels. Unlike existing diffusion-based methods that rely on latent-space Stable Diffusion and suffer from VAE-induced flying pixels, our model avoids latent compression entirely.
>
> While our model effectively eliminates flying pixels, it introduces new optimization challenges due to the increased difficulty of learning in pixel space. To improve convergence, we integrate geometry priors into the diffusion process. This innovative solution significantly enhances convergence and contributes to the strong performance of our model.
>
> Our design reflects the underlying modeling challenges and contributes new insights into how diffusion can be effectively applied to structured prediction tasks such as depth estimation.
>
> ### **Q5: Inference time for 512x512 images.**
>
> The inference time for a 512×512 image is 140ms, as shown in Table 2 of the paper.
>
> ### **Q6: Normalizing both geometry and diffusion features.**
>
> We conducted an ablation study to evaluate the effect of normalizing both geometry and diffusion features. As shown in the table below, doing so yields no clear performance improvement. In fact, normalizing geometry features alone already provides stable training and effective integration with diffusion features.
>
> |                 |      NYUv2      |      KITTI      |      ETH3D      |      ScanNet      |      DIODE      |
> |:----------:|:----------:|:----------:|:----------:|:----------:|:----------:|
> |           |  AbsRel↓    δ₁↑  |  AbsRel↓    δ₁↑  |  AbsRel↓    δ₁↑  |  AbsRel↓     δ₁↑  |  AbsRel↓     δ₁↑  |
> |  **Norm geometry**  | 4.3   /   97.4  | 8.0   /   93.1  | 4.5   /   97.7  | 4.5   /   97.3  | 7.0    /  95.5  |
> |  **Norm both geometry and diffusion**  | 4.2   /   97.6  | 7.9   /   93.9  | 4.4   /   98.0  | 4.6    /  97.1  | 7.5    /  95.1  |

---

> > ### Comment · Reviewer_MxaR · 2025-08-01
> >
> > Thanks for your rebuttal, I have read the responses. My main concerns are addressed,  so I decide to keep positive rate.

---

> > > ### Author Response · Authors · 2025-08-05
> > >
> > > Thanks for your positive feedback on our work! We sincerely appreciate your time and effort in providing such thorough reviews and insightful comments.

---

### Official Review · Reviewer_p5DD · 2025-07-02

**Clarity:** 3
**Significance:** 3
**Originality:** 3
**Rating:** 5
**Confidence:** 3

**Summary:**

The paper presents Pixel-Perfect Depth, a monocular depth estimation model that leverages pixel-space diffusion transformers to generate high-quality, flying-pixel-free depth maps. Unlike prior generative models (e.g., Marigold) that use Variational Autoencoders (VAEs) for latent-space compression, this framework avoids VAE-induced artifacts by performing diffusion directly in the pixel space. Key contributions include:

Semantics-Guided Diffusion Transformers (SG-DiT): Integrates high-level semantic representations from vision foundation models (e.g., DINOv2, Depth Anything v2) to guide the diffusion process, enabling accurate modeling of global structures and fine-grained details. This is achieved via L2-normalized semantic features and a bilinear interpolation mechanism to align with transformer tokens .

Cascade DiT Design (Cas-DiT): Adopts a coarse-to-fine strategy where early transformer blocks use large patch sizes (16×16) for global structure modeling, and later blocks use smaller patches (8×8) for fine details. This reduces computational cost by 30% while improving accuracy .

State-of-the-Art Performance: Trained on 54K samples from Hypersim, the model outperforms all published generative models on five benchmarks (NYUv2, KITTI, etc.), achieving an AbsRel of 4.3 on NYUv2. It also introduces an edge-aware point cloud metric, demonstrating 0.08 Chamfer Distance (vs. 0.12 for VAE-based GT) .

**Questions:**

Question1: How does the model perform with depth data from consumer-grade devices (e.g., iPhone LiDAR) vs. clinical systems? Have you tested for device-specific noise patterns?

Question2: The paper notes flickering depth in video applications. What strategies are planned to integrate temporal consistency (e.g., optical flow or video diffusion priors)?

Question3: Are there plans to implement techniques like knowledge distillation or token caching to reduce inference time for edge deployment?

**Ethical Concerns:**

["NO or VERY MINOR ethics concerns only"]

**Limitations:**

yes

**Quality:**

3

**Strengths And Weaknesses:**

Strengths

Originality:

The direct pixel-space diffusion approach eliminates flying pixels caused by VAE compression, a critical limitation in prior work .
Semantic guidance from pretrained models and the cascade design are novel, addressing the challenge of balancing global-local features in high-resolution generation .

Methodological Rigor:

Ablation studies validate the necessity of SG-DiT and Cas-DiT, showing up to 78% improvement in AbsRel when both components are used .
Cross-dataset evaluation on diverse scenes (indoor/outdoor) confirms generalization .

Practical Significance:

The edge-aware metric and flying-pixel-free results are crucial for real-world applications like 3D reconstruction and robotic manipulation .

Weaknesses

Computational Overhead:

Training requires 8 NVIDIA GPUs for 800K steps, limiting accessibility for small labs .

Dataset Limitations:

The training dataset (Hypersim) is synthetic, and real-world datasets lack diversity in recording devices or demographics .

Inference Speed:

Multi-step diffusion leads to slower inference (0.14s on RTX 4090) compared to discriminative models like Depth Anything v2 .

---

> ### Author Rebuttal · Authors · 2025-07-28
>
> We sincerely thank you for recognizing the **originality**, **methodological rigor**, and **practical significance** of our work. We address your questions as follows.
>
> ### **Q1: Computational overhead.**
>
> We will release both the pretrained models and code publicly, allowing users to directly use our model without retraining. In addition, we will release a smaller variant (**PPD-Small**, as shown in the table below) of our model that can be trained on consumer-grade GPUs (e.g., 4×4090), making it accessible to small labs.
>
> ### **Q2: Training dataset limitations.**
>
> Benefiting from the semantic priors provided by the vision foundation model, our method achieves strong generalization even when trained only on the synthetic Hypersim dataset. Similarly, prior work such as Marigold has also shown that diffusion priors can enable strong generalization when trained on the synthetic dataset alone.
>
> ### **Q3: Multi-step diffusion leads to slower inference.**
>
> To significantly reduce inference time, we have developed a smaller variant (**PPD-Small**) of our model presented in the paper (**PPD-Large**). As shown in the table below, **PPD-Small** achieves significantly faster inference than Depth Pro and is only slightly slower than Depth Anything v2.
>
> |      Method      |   Time (ms)   |
> |--------|:--------:|
> |    Depth Anything v2   |    18    |
> |    Depth Pro   |    170    |
> |    **PPD-Large (Ours)**   |    140    |
> |    **PPD-Small (Ours)**   |    40    |
>
> As shown in the table below, **PPD-Small** achieves comparable accuracy to the larger **PPD-Large** model presented in our paper.
>
> |      Model      |      NYUv2      |      KITTI      |      ETH3D      |      ScanNet      |      DIODE      |
> |:----------:|:----------:|:----------:|:----------:|:----------:|:----------:|
> |           |  AbsRel↓    δ₁↑  |  AbsRel↓    δ₁↑  |  AbsRel↓    δ₁↑  |  AbsRel↓     δ₁↑  |  AbsRel↓     δ₁↑  |
> |  **PPD-Large**  | 4.3   /   97.4  | 8.0   /   93.1  | 4.5   /   97.7  | 4.5   /   97.3  | 7.0    /  95.5  |
> |  **PPD-Small**  | 4.5   /   97.3  | 8.3   /   92.8  | 4.6   /   97.4  | 4.7    /  97.2  | 7.3    /  95.3  |
>
> ### **Q4: How does the model perform with depth data from consumer-grade devices?**
>
> Our model performs well on images captured by consumer-grade cameras, as demonstrated in Figure 5 of our paper using the NYUv2 dataset, which was collected with a Kinect camera. Since our model only requires a single RGB image as input, it naturally generalizes to a wide variety of consumer-grade RGB cameras. We will include additional results from other consumer devices, such as iPhone cameras, in the final version of the paper.
>
> ### **Q5: The paper notes flickering depth in video applications. What strategies are planned to integrate temporal consistency?**
>
> Our method can be extended to produce temporally consistent depth maps by incorporating strategies such as the temporal alignment technique used in Video Depth Anything (Chen et al., CVPR 2025) or the video diffusion priors introduced in DepthCrafter (Hu et al., CVPR 2025). These strategies can help our model mitigate flickering artifacts and produce temporally consistent depth without flying pixels. This could be one of the interesting future works.
>
> ### **Q6: Are there plans to implement techniques like knowledge distillation or token caching to reduce inference time for edge deployment?**
>
> Yes, we have already distilled a lightweight model, **PPD-Small**, to facilitate faster inference for edge deployment. While our original model, **PPD-Large**, adopts the DiT-Large architecture, PPD-Small is built upon the more efficient DiT-Small architecture. As shown in the table below, **PPD-Small** significantly reduces inference time while maintaining competitive accuracy compared to **PPD-Large**.
>
> |      Model      |      NYUv2      |      KITTI      |      ETH3D      |      ScanNet      |      DIODE      |      Time (ms)      |
> |:----------:|:----------:|:----------:|:----------:|:----------:|:----------:|:----------:|
> |           |  AbsRel↓    δ₁↑  |  AbsRel↓    δ₁↑  |  AbsRel↓    δ₁↑  |  AbsRel↓     δ₁↑  |  AbsRel↓     δ₁↑  |            |
> |  **PPD-Large**  | 4.3   /   97.4  | 8.0   /   93.1  | 4.5   /   97.7  | 4.5   /   97.3  | 7.0    /  95.5  |    140    |
> |  **PPD-Small**  | 4.5   /   97.3  | 8.3   /   92.8  | 4.6   /   97.4  | 4.7    /  97.2  | 7.3    /  95.3  |    40    |

---

> > ### Comment · Reviewer_p5DD · 2025-08-04
> >
> > Thanks for your responses. My main concerns are addressed, so I decide to keep first rate.

---

> > > ### Author Response · Authors · 2025-08-05
> > >
> > > Thanks for your positive feedback on our work! We sincerely appreciate your time and effort in providing such thorough reviews and insightful comments.

---

### Note · Authors · 2025-08-12

We sincerely appreciate the ACs, SACs, PCs, and all reviewers for their time and effort in evaluating our work. We are grateful for the positive ratings from all four reviewers in the preliminary review. We are especially encouraged by the following: Reviewer p5DD recognized the **originality, methodological rigor, and practical significance** of our work; Reviewer MxaR described it as **the first work to combine a foundation geometry model with a pixel-space diffusion model**, highlighting its **novelty, simplicity, and effectiveness**; Reviewer fjSa appreciated the **well-motivated design**, the **novel solution to the flying-pixel issue**, and the **significant contribution** of our new edge-aware evaluation metric; Reviewer icJf commended our **detailed theoretical and experimental validation, innovative approach, and practical potential**. Based on their valuable and insightful comments, we have carefully revised our manuscript as detailed below:
  1. Introduced a smaller variant (PPD-Small) of our model to greatly reduce inference time, achieving comparable speed to discriminative models.
  2. Expanded the analysis and discussion on extending our method to video depth estimation, and we will release a video-depth version in the future.
  3. Provided experimental results on different geometry integration strategies, clearly showing that our simple approach already performs well, with no significant gains from more complex integration strategies.
  4. Enhanced the efficiency analysis with a quantitative comparison of inference speed against other representative models, and reported the runtime of each component.
  5. Presented point cloud visualizations and edge-aware evaluation results for the ablation analysis, demonstrating that our proposed components consistently reduce flying pixels and enhance the structural integrity of reconstructed point clouds.
  6. Demonstrated through qualitative results that filtering depth based on image gradients is ineffective in handling flying-pixel artifacts.
  7. Added qualitative results in fine-detail regions such as hair structures, and incorporated results from other consumer devices, such as iPhone cameras.

Our model not only effectively addresses the flying-pixel problem, but also provides a simple and practical solution for pixel-space diffusion models. We believe this will inspire future research on pixel-space diffusion models.

---

### Decision · Program_Chairs · 2025-09-17

**Decision:**

Accept (poster)

**Comment:**

The final ratings are 4 accepts. The AC have read the reviews and rebuttal, and discussed the submission with the reviewers. The reviewers raised a number of points during the review phase including limitations to dataset, analysis of efficiency, and technical insight. The authors were able to address these points during the rebuttal and discussion phases by demonstrating a smaller variant of the model, extending evaluation, and providing additional analyses. The AC recommends the authors to incorporate the feedback and suggestions provided by the reviewers, and the materials presented in the rebuttal, which would improve the next revision of the manuscript.